# Aragonite lithium/magnesium as an indicator of calcification media saturation state in marine calcifiers

Cristina Castillo Alvarez[1,2], Edmund Hathorne[3], Matthieu Clog[4], Adrian Finch[1], Roland Kröger [5], Kirsty Penkman [6] & Nicola Allison [1,2] ✉

Marine calcifiers support ecosystem services, including shell fisheries and coral reefs. Constraining the saturation state of the calcification media of these organisms is essential to understand the response of biomineralisation to environmental change. Here we synthesise aragonite over variable pH, saturation state, temperature, and in the presence of simple biomolecules. We show that the lithium/magnesium distribution coefficient, relating aragonite and precipitation fluid compositions, is significantly affected by precipitation rate but not by temperature or pH. Precipitation rate reflects saturation state and temperature, so lithium/magnesium of biogenic aragonite can be used to calculate mineral precipitation rate and, if the precipitation temperature is known, to reconstruct calcification medium saturation state. Applying the distribution coefficients to a published calcifier dataset indicates that calcification media saturation state is *ca.* 9 to 13 at 18–30 °C and *ca.* 6 to 10 at 10–18 °C. Coral calcification media saturation state varies between ocean sites, species, and reef zones.

Calcification is the production of $CaCO_3$ structures, e.g., shells, plates, and skeletons, by organisms including corals, molluscs, and foraminifera. This process provides organisms with tissue support and protection from predators and the environment, constructs habitat spaces for other organisms (e.g., coral reefs), and plays an important role in carbon and calcium biogeochemical cycles[1]. Inorganic precipitation rates of $CaCO_3$ reflect seawater $CaCO_3$ saturation state, $\Omega$, which is a function of seawater $[Ca^{2+}]$ and $[CO_3^{2-}]$[2] and the presence of other ions, e.g., $Mg^{2+}$, $SO_4^{2-}$, and $PO_4^{3-}$[3]. Biogenic $CaCO_3$ structures are formed from calcification media hosted intracellularly[4–6] and extracellularly[6,7]. The calcification media are typically sourced from seawater[4,8], but many organisms elevate media pH[7,9,10], shifting the dissolved inorganic carbon (DIC) equilibrium in favour of $CO_3^{2-}$ and promoting the formation of $CaCO_3$. Therefore, fully characterising calcification media $\Omega$ ($\Omega_{CM}$) is essential to understand biomineralisation processes and to predict the effects of future environmental change on marine ecosystem services. Aragonite B/Ca has been used to infer calcification media $[CO_3^{2-}]$[11] in combination with aragonite $\delta^{11}B$ used to indicate calcification media pH[12]. However, a recent study indicates that aragonite B/Ca is not influenced by precipitating fluid $[CO_3^{2-}]$ at the pH and DIC conditions of tropical coral calcification sites[13], while $\delta^{11}B$ estimates of coral calcification media pH

are substantially higher than estimates from the pH-sensitive dye SNARF-1 in the extracellular calcification site[14].

Here, we determine how $\Omega_{Ar}$ (aragonite saturation state), temperature, pH, and mineral growth rate (R) influence Mg and Li partitioning in aragonite, and we explore Li/Mg as a proxy of aragonite growth rate and $\Omega_{Ar}$. We precipitate aragonite in vitro from artificial seawater over a range of pH and aragonite saturation states ($\Omega_{Ar}$), including those inferred to occur in coral calcification media[13]. We combine a pH stat titrator, a $Ca^{2+}$ dosing system, and a gas control apparatus to ensure that pH, $\Omega$, and $[Ca^{2+}]$ remain essentially constant within each precipitation. $[CO_3^{2-}]$ and pH covary between precipitations at atmospheric $CO_2$[13], so we conduct experiments under different $CO_2$ atmospheres to deconvolve the influences of pH and $[CO_3^{2-}]$ on Mg and Li partitioning. Calcification is affected by organic matrices at the calcification site, which influence mineral deposition[15], so we also test the effect of common biomineral amino acids on aragonite Li and Mg incorporation.

## Results and Discussion
### pH and growth rate influences on $D_{Li/Mg}$

Firstly, we analyse a suite of aragonite samples precipitated at 25 °C, salinity = 35 in experiments which deconvolved the influences of pH, $[CO_3^{2-}]$, and

[1]School of Earth and Environmental Sciences, University of St Andrews, St Andrews, UK. [2]Scottish Ocean Institute, University of St Andrews, St Andrews, UK. [3]GEOMAR, Helmholtz Centre for Ocean Research Kiel, Kiel, Germany. [4]SUERC, University of Glasgow, Glasgow, UK. [5]School of Physics, Engineering and Technology, University of York, York, UK. [6]Department of Chemistry, University of York, York, UK. ✉e-mail: na9@st-andrews.ac.uk

$[HCO_3^-]$ on aragonite precipitation rate[16]. We analyse a subset of the precipitations previously reported[16] and supplement these with a small number of additional precipitations. The DIC conditions and aragonite precipitation rates of these experiments are summarised in Fig. 1a and b, respectively.

Seawater and aragonite geochemistry are determined by ICP-OES and ICP-MS, respectively. Aragonite distribution coefficients are calculated as:

$$D_{Me/Ca} = Me/Ca_{aragonite} / Me/Ca_{solution}, \text{ where Me is Li or Mg} \quad (1)$$

$$D_{Li/Mg} = Li/Mg_{aragonite} / Li/Mg_{solution} \quad (2)$$

where Me/Ca and Li/Mg are measured in mol mol$^{-1}$.

$D_{Mg/Ca}$, $D_{Li/Ca}$, and $D_{Li/Mg}$ are significantly related to aragonite precipitation rate (Fig. 2a-c, Table 1, equations 3 to 5, respectively), and, thereby to $\Omega_{Ar}$ (Fig. 1b). $D_{Mg/Ca}$ and $D_{Li/Ca}$, are positively related to growth, as reported previously in aragonite[17-19] and calcite[20-22]. $D_{Mg/Ca}$, $D_{Li/Ca}$, and $D_{Li/Mg}$ show no dependence on seawater pH (Fig. 2a-c, equations 3 to 5 in Table 1). The incorporation of trace elements in $CaCO_3$ and the $CaCO_3$ crystallisation process itself are poorly understood[23]. $CaCO_3$ trace element chemistry is influenced by the attachment/detachment rates of trace element ions to the mineral compared to the behaviour of host ions[24], by the diffusion rates of ions through the mineral: fluid boundary[2,3] and by the formation of precursor phases e.g. amorphous calcium carbonate[25].

Solid-state diffusion in the newly formed mineral surface[26] is likely to be too slow in carbonates to be significant at environmental temperatures[24]. In seawater, alkali and alkaline earth metals predominantly exist as hydrated cations[27]. Li$^+$ may also complex with OH$^-$ or $CO_3^{2-}$ to form hydrated complexes[28], the abundance of which is pH dependent[29]. Both Mg$^{2+}$ and Li$^+$ have smaller ionic radii that Ca$^{2+}$ [30]. Incorporation of Mg$^{2+}$ in aragonite either occurs by substitution (for Ca$^{2+}$) and relaxation of the lattice structure or by accommodation of nanodomains[31]. Li$^+$ is incorporated in aragonite at adjacent substitutional and interstitial sites i.e., two Li$^+$ ions occupy one Ca$^{2+}$ site[32]. Although there is evidence of LiHCO$_3$ incorporation in calcite[21,22], we observe no effect of pH (Table 1) or fluid [HCO$_3^-$] (Fig. S2) on $D_{Li/Ca}$ to support this hypothesis in aragonite. Relatively large variations occur in

$D_{Li/Ca}$ between different pH treatments at high aragonite growth rates (Fig. 2b) which warrant future investigation. However, fluid $\Omega_{Ar}$ in these fast growth rate experiments ($\Omega_{Ar} \geq 17$) is considerably higher than observed in the calcification media of tropical corals cultured at ambient [CO$_2$] where $\Omega_{Ar} \approx 12$[10], suggesting this observation is not relevant to marine calcifiers. $D_{Li/Ca}$ and $D_{Mg/Ca}$ are «1 (Fig. 2a,b), indicating that both Li$^+$ and Mg$^{2+}$ are much less likely to be incorporated in the lattice than Ca$^{2+}$.

## Temperature influences on $D_{Me/Ca}$ and $D_{Li/Mg}$

Secondly, we analyse aragonite synthesised over varying $\Omega_{Ar}$ and temperature (Fig. 1c), also at salinity = 35. Both $\Omega_{Ar}$ and temperature affect aragonite precipitation rate significantly (Fig. 1d, equation 6 in Table 1). These precipitations were conducted at ambient $CO_2$ and we observe a broad positive relationship between seawater pH and $\Omega_{Ar}$, over the sample suite (Fig. S1). As we have already determined that pH does not affect $D_{Mg/Ca}$, $D_{Li/Ca}$, and $D_{Li/Mg}$, we combine the data from these temperature precipitations with those reported in the previous section to analyse the effects of temperature and growth rate on $D_{Mg/Ca}$, $D_{Li/Ca}$, and $D_{Li/Mg}$.

$D_{Mg/Ca}$ and $D_{Li/Ca}$ are significantly affected by both temperature and aragonite precipitation rate (Fig. 2d,e, equations 7 and 8 in Table 1), however $D_{Li/Mg}$ reflects precipitation rate only (Fig. 2f, equations 9 and 10 in Table 1). $D_{Li/Mg}$ is independent of temperature as $D_{Li/Ca}$ and $D_{Mg/Ca}$ exhibit similar sensitivity to temperature but not to growth rate. As an example, increasing the temperature from 25 to 30°C increases $D_{Mg/Ca}$ and $D_{Li/Ca}$ by 15 and 18%, respectively, at a log precipitation rate of 3.2. In contrast, increasing the log precipitation rate from 2.4 to 3.4 at 25 °C increases $D_{Mg/Ca}$ by 180% and $D_{Li/Ca}$ by only 20%.

Although log precipitation rate and temperature generated similar trends in both $D_{Mg/Ca}$ and $D_{Li/Ca}$ in previous aragonite precipitation studies[17,19], $D_{Mg/Ca}$ and $D_{Li/Ca}$ in the present study are considerably higher than in these previous reports (Fig. S3a, b). We observe an inverse relationship between precipitation rate and $D_{Li/Mg}$, in contrast to the positive relationship reported by Brazier et al.[17] (Fig. S3c). This previous study used simple solutions (270 mM NaCl, 25 mM MgCl$_2$), with [Li] more than x500, that used in our experiments. This generated aragonite [Li] ~x100 higher[17]

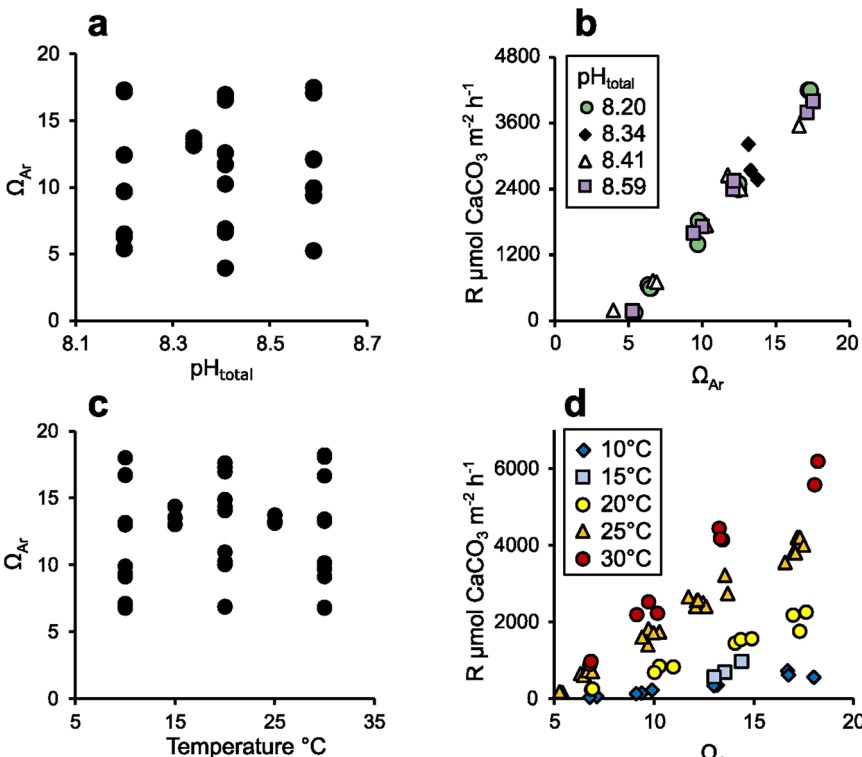

**Fig. 1 | Summary of the solution conditions and precipitation rates in the synthetic aragonite experiments. a** solution conditions and **b** aragonite precipitation rates (R) in 25 °C experiments, and **c** solution conditions and **d** aragonite precipitation rates at variable temperature. Typical errors in pH. $\Omega_{Ar}$, temperature within each precipitation is estimated to be 0.003 pH units, 0.04 to 0.16 $\Omega_{Ar}$, and 0.04 °C, while the error in precipitation rate is estimated to be ~3% (see methods). In all cases, these errors are smaller than the symbols used.

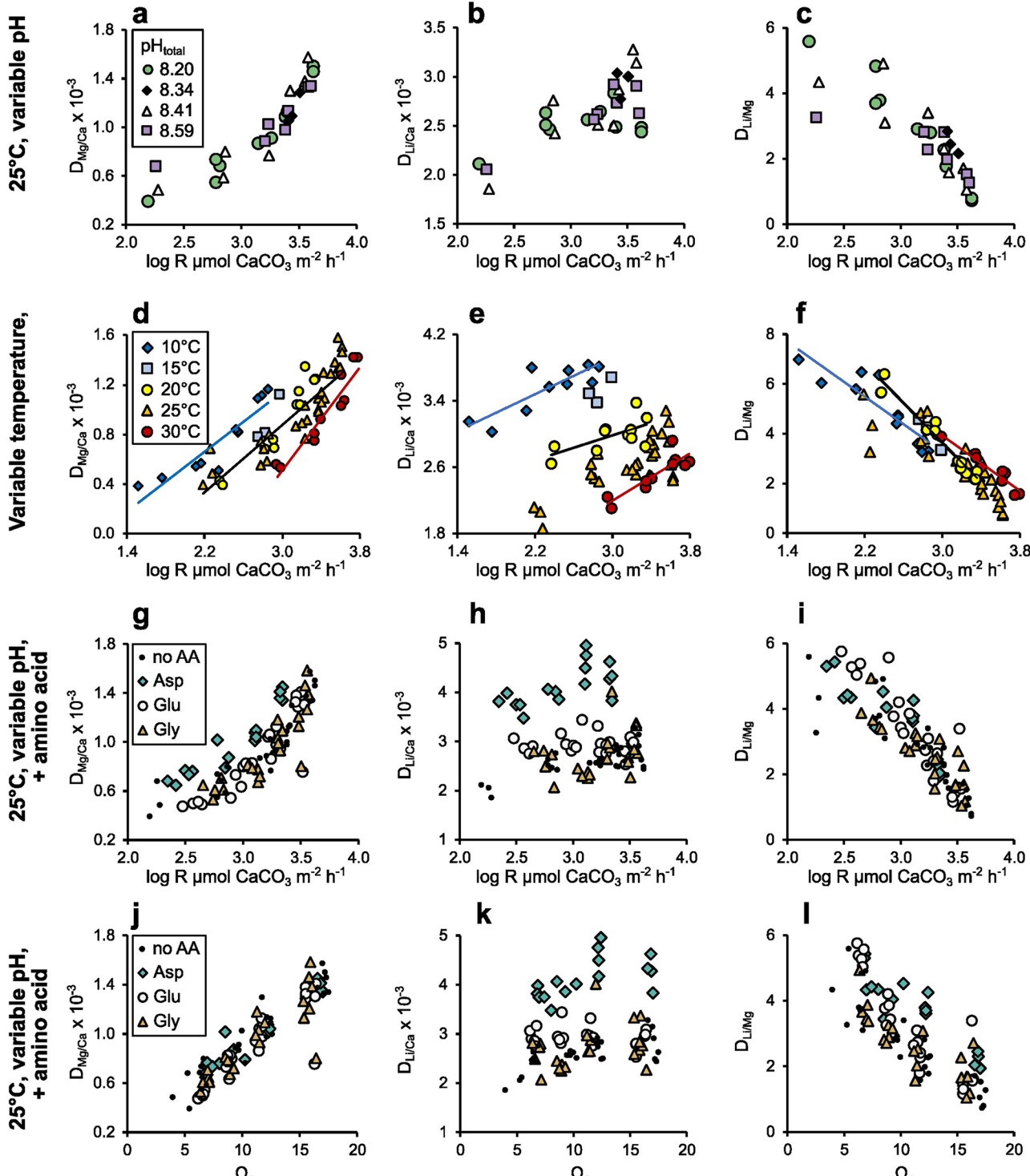

**Fig. 2 | Influences on $D_{Mg/Ca}$, $D_{Li/Ca}$ and $D_{Li/Mg}$.** Relationships between $D_{Mg/Ca}$, $D_{Li/Ca}$ and $D_{Li/Mg}$ and aragonite precipitation rate (R) or $\Omega_{Ar}$ as a function of **a–c** pH at 25 °C, **d–f** variable temperature and **g–i** in the presence of amino acids at 25 °C. **j–l** Relationships between $\Omega_{Ar}$ and $D_{Mg/Ca}$, $D_{Li/Ca}$, and $D_{Li/Mg}$ in the presence of amino acids. Best fit linear relationships are fitted to data at 10, 20, and 30 °C in **d–f**. No AA = no amino acid, Asp = aspartic acid, Glu = glutamic acid, Gly = glycine in **g–l**. Errors in $D_{Mg/Ca}$, $D_{Li/Ca}$ and $D_{Li/Mg}$ (1.4%, 2.6% and 3.1%, respectively), precipitation rate and $\Omega_{Ar}$ are smaller than the symbols used (see methods).

than observed in the present study or in marine biominerals[33]. Trace element ions, which either substitute for $Ca^{2+}$ in the $CaCO_3$ lattice, or are hosted as interstitial ions between crystal lattice sites, create lattice distortions[31,34] and the attachment/detachment of trace elements is influenced by the presence of other non-host ions on the crystal surface[24,35]. Further work is required to identify how aragonite [Li] influences further Li

incorporation. However, we observe good agreement in $D_{Li/Ca}$ and $D_{Li/Mg}$ (Fig. 2e and f) between experiments conducted from 2 different batches of seawater with [Li] approximately equal to that of seawater or with [Li] approximately half of this (Table S1 and Table S2). This suggests that minor variations in seawater [Li] have a limited effect on Li partitioning in aragonite.

**Table 1 | Equations describing $D_{Me/Me}$ as a function of aragonite growth rate (R, µmol m$^{-2}$ h$^{-1}$) and pH (total scale) in precipitations conducted at 25 °C and as a function of log precipitation rate and temperature (T) in precipitations conducted over variable temperature**

| No. | Equation | r² | p value | | |
|---|---|---|---|---|---|
| | **DMe/Me as a function of log precipitation rate (R) and pH at 25 °C (n = 28)** | | *rate* | *pH* | *intercept* |
| 3 | $D_{Mg/Ca} = 6.79\ (\pm0.74) \times 10^{-4} \log R + 7.56\ (\pm19.8) \times 10^{-5} pH - 1.79\ (\pm1.65) \times 10^{-2}$ | 0.80 | **5.4 × 10⁻⁹** | 0.71 | 0.29 |
| 4 | $D_{Li/Ca} = 5.06\ (\pm1.05) \times 10^{-4} \log R + 1.59\ (\pm2.83) \times 10^{-4} pH - 3.28\ (\pm23.5) \times 10^{-4}$ | 0.48 | **8.5 × 10⁻⁵** | 0.58 | 0.89 |
| 5 | $D_{Li/Mg} = -2.56\ (\pm0.32) \log R - 0.688\ (\pm0.862) pH + 16.3\ (\pm7.2)$ | 0.75 | **6.3 × 10⁻⁸** | 0.45 | 0.033 |
| | **Log precipitation rate (R) as a function of $\Omega_{Ar}$ and T (n = 63)** | | $\Omega_{Ar}$ | *T* | *Intercept* |
| 6 | $\text{Log}R = 8.77\ (\pm0.50) \times 10^{-2} \Omega Ar + 5.53\ (\pm0.31) \times 10^{-2} T + 0.781\ (\pm0.093)$ | 0.91 | **2.6 × 10⁻²⁵** | **1.0 × 10⁻²⁵** | **1.2 × 10⁻¹¹** |
| | **DMe/Me as a function of log precipitation rate (R) and T (n = 63)** | | *rate* | *T* | *intercept* |
| 7 | $D_{Mg/Ca} = 7.19\ (\pm0.48) \times 10^{-4} \log R - 2.56\ (\pm0.38) \times 10^{-5} T - 6.88\ (\pm1.09) \times 10^{-4}$ | 0.80 | **3.6 × 10⁻²²** | **7.0 × 10⁻⁹** | **3.7 × 10⁻⁸** |
| 8 | $D_{Li/Ca} = 5.44\ (\pm0.65) \times 10^{-4} \log R - 8.92\ (\pm0.52) \times 10^{-5} T + 3.19\ (\pm0.15) \times 10^{-3}$ | 0.83 | **1.0 × 10⁻¹¹** | **6.9 × 10⁻²⁵** | **8.3 × 10⁻³⁰** |
| 9 | $D_{Li/Mg} = -2.75\ (\pm0.21) \log R + 1.03\ (\pm1.67) \times 10^{-2} T + 11.4\ (\pm0.5)$ | 0.82 | **2.7 × 10⁻¹⁹** | 0.62 | **3.2 × 10⁻³²** |
| | **$D_{Li/Mg}$ as a function of log precipitation rate (R) (n = 63)** | | *rate* | | *intercept* |
| 10 | $D_{Li/Mg} = -2.66\ (\pm0.15) \log R + 11.4\ (\pm0.5)$ | 0.83 | **4.2 × 10⁻²⁵** | | **9.2 × 10⁻³³** |

The equations for aragonite precipitation rate as a function of $\Omega_{Ar}$ and T and for $D_{Li/Mg}$ as a function of precipitation rate only, are also included. Standard errors of equation coefficients and intercepts are included in brackets. Coefficients of determination (r²) and p values for each equation coefficient are shown.
Significant p values are highlighted in bold.

**Table 2 | One way ANCOVA p values comparing relationships between $D_{Me/Me}$ and log aragonite precipitation rate or $\Omega_{Ar}$ in experiments performed at 25 °C in the presence and absence of aspartic acid, glutamic acid and glycine**

| | p (equal means) | p (equal slopes) | Significant difference from control |
|---|---|---|---|
| **Log aragonite precipitation rate** | | | |
| $D_{Mg/Ca}$ | **8.6 × 10⁻⁷** | 0.38 | Mean aspartic acid > control |
| $D_{Li/Ca}$ | **1.8 × 10⁻²⁷** | 0.039 | Mean aspartic acid, glutamic acid > control |
| $D_{Li/Mg}$ | 0.22 | **0.0013** | Slope glutamic acid ≠ control |
| $\Omega_{Ar}$ | | | |
| $D_{Mg/Ca}$ | 0.082 | 0.90 | ns |
| $D_{Li/Ca}$ | **2.8 × 10⁻²⁵** | 0.058 | Mean aspartic acid, glutamic acid > control |
| $D_{Li/Mg}$ | **3.1 × 10⁻⁷** | 0.034 | Mean aspartic acid, glutamic acid > control |

ANCOVA tests for equal means (after correcting for variance) and equal slopes between populations. To avoid type 1 errors (generated by rejecting a true null hypothesis) we apply a conservative Bonferroni correction to the p-value. We ran a total of 6 tests to compare relationships between Me/Me and precipitation rate or $\Omega_{Ar}$ between the control and each amino acid treatment, and we calculated an adjusted α value of 0.020 (i.e., the original α value, 0.05, divided by the square root of the total number of tests). We conclude that relationships vary significantly between populations when p ≤ 0.020. Significant p-values are highlighted in bold. Amino acid treatments that are significantly different from the control are noted. ns = not significant.

### Amino acid effects on $D_{Me/Ca}$ and $D_{Li/Mg}$

Finally, we analyse aragonite samples precipitated at 25°C over a range of pH and $\Omega_{Ar}$ (as in Fig. 1a) and in the presence of 2 mM of 3 amino acids (aspartic acid, glutamic acid and glycine), which are abundant in coral skeletons[36–38] and mollusc shells[39–41]. Precipitation rates of these samples[16], show that all amino acids inhibit aragonite precipitation, with aspartic acid and glycine the most and least effective inhibitors, respectively.

Aspartic and glutamic acids significantly increase $D_{Li/Ca}$ as a function of both precipitation rate and $\Omega_{Ar}$ (Fig. 2h, k, Table 2). Aspartic acid increases $D_{Mg/Ca}$ as a function of precipitation rate but not $\Omega_{Ar}$ (Fig. 2g, j) and increases $D_{Li/Mg}$ as a function of $\Omega_{Ar}$ but not precipitation rate (Fig. 2i, l). Amino acids complex $Mg^{2+}$ and $Ca^{2+}$ in solution[42] and create lattice distortions when incorporated into calcite[43]. Both processes could alter the relative rates of trace element and $Ca^{2+}$ adsorption to the mineral surface. Although our study shows that amino acids influence the Li/Mg versus $\Omega_{Ar}$ relationship, we note that the seawater [amino acid] tested here (2 mM) far exceeds that likely to occur at organism calcification sites. Intra-crystalline [aspartic acid] is 0.5 to 1.5 nmol mg$^{-1}$ in coral skeletons[37] and ≤1 nmol mg$^{-1}$ in mollusc shells[44], but is >13 nmol mg$^{-1}$ in synthetic aragonite precipitated under the conditions used here[45]. In vitro precipitations with 100 µM aspartic acid produce aragonite which has a comparable [aspartic acid] i.e., 0.8 nmol mg$^{-1}$[45], to biominerals. Aragonite precipitation rates are reduced

by ~12% by 100 µM aspartic acid[46] and any influence of this on aragonite Li/Mg will be small (Fig. 2i).

### Applications to biogenic aragonite

Although aragonite Li/Mg has been identified as a palaeothermometer[47,48], our study shows that $D_{Li/Mg}$ is not significantly affected by temperature but is inversely related to aragonite precipitation rate. Aragonite precipitation rates in vitro reflect temperature and $\Omega_{Ar}$[2,13] (Fig. 1f) and the presence of biomolecules[16,37,45,46]. We consider that biomolecules are unlikely to significantly affect the precipitation rate versus aragonite Li/Mg relationship in marine calcifiers (Fig. 2i). We conclude that biogenic aragonite Li/Mg is a proxy for mineral precipitation rate. If the precipitation temperature is known i.e., in cultured organisms or those collected from known temperature environments, then aragonite Li/Mg can be used to reconstruct calcification media $\Omega_{Ar}$.

To demonstrate these applications, we utilise a composite dataset of Li/Mg in modern corals and the aragonitic foraminifera, *Hoeglundina elegans*, that were collected from known temperature environments, reproduced in Fig. 3a[33]. We rewrite equation 10 (Table 1) as:

$$\text{Log R} = (D_{Li/Mg} - 11.4)/-2.66 \qquad (3)$$

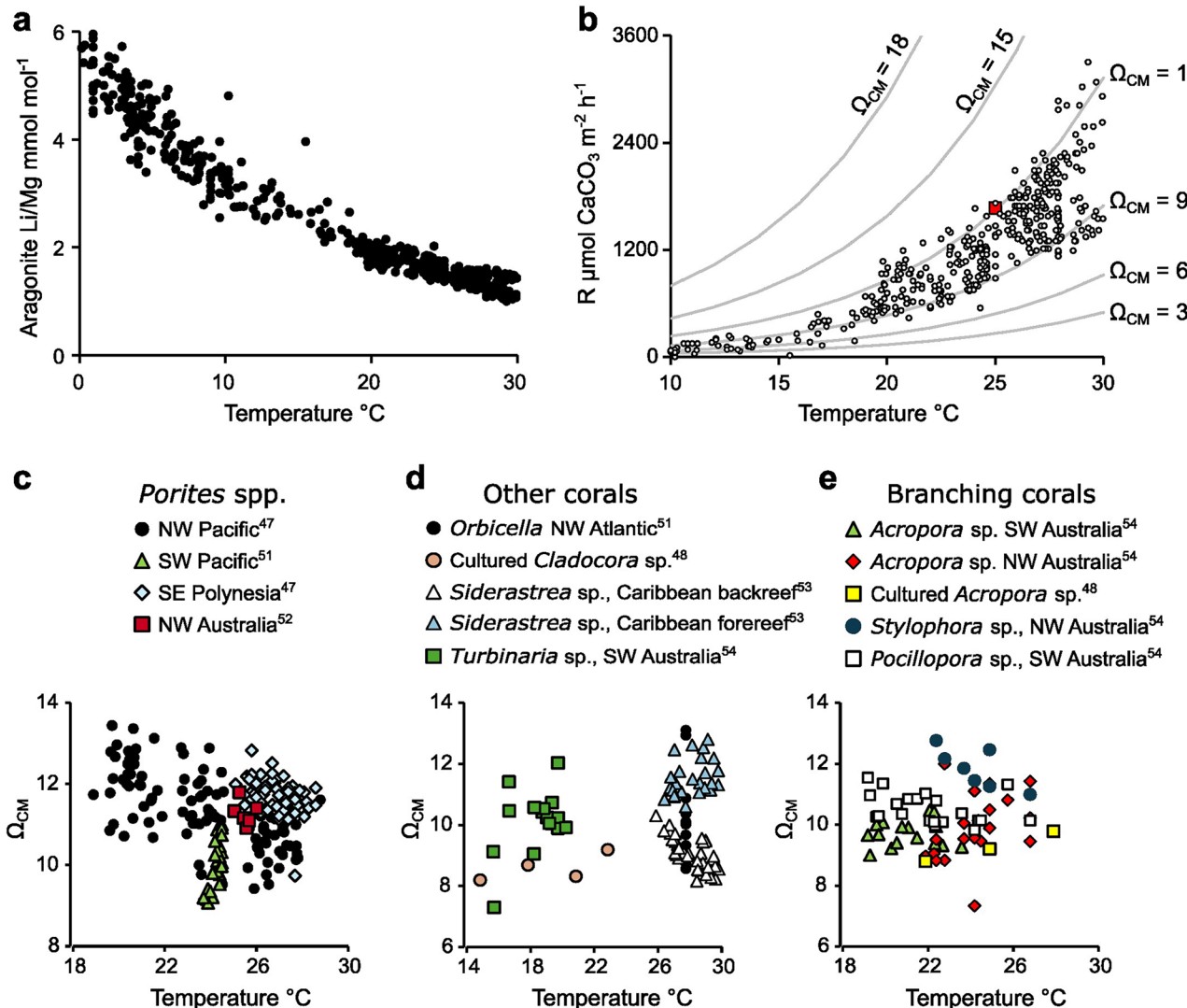

**Fig. 3 | Applications of $D_{Li/Mg}$ in biogenic aragonite. a** Reproduction of a composite Li/Mg dataset from aragonitic corals and foraminifera as a function of precipitation temperature[33], **b** estimates of aragonite precipitation rates (R) from Li/Mg in the composite dataset for samples from temperatures 10 to 30 °C using Eq. 10, Table 1. Contours show aragonite precipitation rates observed in synthetic aragonite in this study (Eq. 6, Table 1), and the red dot indicates the measured $\Omega_{Ar}$ in the extracellular calcification media of the branching coral *Stylophora pistillata*, cultured at 25 °C[10]. Estimated $\Omega_{CM}$ as a function of known environment temperature and published Li/Mg (using Eq. 10, Table 1) in **c** massive *Porites* spp., **d** other Symbiodinicaea hosting corals and **e** branching corals.

We calculate $D_{Li/Mg}$ from Li/Mg$_{aragonite}$ and Li/Mg$_{calcification\ media}$. We consider that Rayleigh fractionation, caused by changes in the relative proportions of different solutes in a fluid reservoir as precipitation occurs[49], has no significant effect on calcification media Li/Mg, as $D_{Li/Mg}$ is very close to 1 (reflecting the similarity of $D_{Mg/Ca}$ and $D_{Li/Ca}$). We assume that calcification media Li/Mg is the same as seawater, i.e. 0.491 mmol mol$^{-1}$ [50]. We use aragonite Li/Mg to calculate precipitation rates (Fig. 3b) from specimens that inhabited environments spanning from 10 to 30 °C (the temperature range studied in the synthetic aragonite experiments). We overlay the plot with contours of precipitation rates at variable $\Omega_{Ar}$ and temperature, calculated from our synthetic aragonite observations (equation 6, Table 1). The estimated biogenic precipitation rates are comparable to those observed in synthetic aragonites precipitated from seawaters with $\Omega_{Ar}$ of ~9 to 13 in organisms growing at 18 to 30 °C, and from seawaters with $\Omega_{Ar}$ of ~6 to 10 in organisms growing at <18 °C. Measurements of $[CO_3^{2-}]$, $[Ca^{2+}]$ and pH, in the extracellular calcification media of the branching coral *Stylophora pistillata*, cultured at 25 °C, yield $\Omega_{Ar} = 12$[10], which is in good agreement with our estimates of calcification media $\Omega_{Ar}$ based on Li/Mg resolved aragonite growth rates (Fig. 3b). This supports our contention that aragonite

Li/Mg reflects aragonite growth rate and $\Omega_{Ar}$, assuming that calcification media Li/Mg approximates that of seawater.

We rearrange equation 6 (Table 1) to:

$$\Omega_{Ar} = (\text{LogR} - 0.0553T - 0.781)/0.0877 \qquad (4)$$

and use Li/Mg estimates of log precipitation rates (R) combined with known precipitation temperature to reconstruct $\Omega_{CM}$ in the specimens or sites with more than 2 analyses (Fig. 3c–e). Our analysis shows that $\Omega_{CM}$ varies in massive *Porites* spp. corals[47,51,52] and is higher in the specimen collected from SE Polynesia[47] compared to the specimen from the NW Pacific[47] at comparable temperatures (Fig. 3c). $\Omega_{CM}$ also varies between other coral species[48,51,53,54] and is considerably higher in a *Siderastrea siderea* coral collected from a fore reef environment compared to an analogue colony collected from the backreef at the same site (Fig. 3d). Both observations are consistent with changes in the $\Omega_{Ar}$ of the seawater sourced for the calcification media. Surface seawater $\Omega_{Ar}$ is higher in the tropical southeastern Pacific compared to the tropical northwestern Pacific[55] and typically higher at the reef front compared to the backreef[56]. Although corals may upregulate

the pH of the calcification media more under increased seawater $pCO_2$ (low $\Omega_{Ar}$), they do not attain the same media pH as observed in corals cultured under ambient $CO_2$ conditions[57] so they cannot completely offset the effects of low seawater $\Omega_{Ar}$. Finally, $\Omega_{CM}$ in *Acropora* spp. are relatively low compared to other branching coral species (Fig. 3e). Calcification in *Acropora pulchra* is less resilient to ocean acidification than in *Pocillopora* spp[58]. and the low $\Omega_{CM}$ may contribute to this sensitivity.

This research explores element partitioning in aragonite precipitated from seawater-based solutions. Further work is required to identify if amorphous calcium carbonate (ACC) occurs as a precursor phase during these experiments and to resolve how ACC transformation affects element partitioning. To broadly compare element partitioning between the synthetic aragonite samples produced here and biogenic aragonite, we calculate $D_{Mg/Ca}$, $D_{Li/Ca}$ and $D_{Li/Mg}$ for the *Porites* spp. coral skeletons illustrated in Fig. 3c. We use the skeletal Mg/Ca and Li/Mg data summarised in Williams et al[33]. and assume that calcification media [Mg], [Li] and [Ca] approximate that of seawater i.e. 52.7 mmol kg$^{-1}$, 25.9 μmol kg$^{-1}$ and 10.3 mmol kg$^{-1}$ [50]. This yields coral skeletal $D_{Mg/Ca} = 0.79 \pm 0.08 \times 10^{-3}$ ($n = 38$), $D_{Li/Ca} = 2.6 \pm 0.2 \times 10^{-3}$ ($n = 38$) and $D_{Li/Mg} = 3.1 \pm 0.4$ ($n = 211$). These values are comparable to the distribution coefficients observed in the synthetic aragonite precipitations reported here (Fig. 2), although we note that the synthetic experiments span a broad range of precipitation rates. We know of no independent estimates of aragonite precipitation rate in coral skeletons. Coral calcification rates are usually reported normalised to the surface area of the coral colony, but aragonite deposition occurs over the existing skeleton in contact with the coral tissue and this represents a much larger surface area[59]. Coral biomineralisation can also be measured as skeletal extension, but this is highly variable within individual skeletons[60]. At this time, a more detailed comparison of distribution coefficients between biogenic and synthetic aragonite is not possible.

Our study shows that aragonite Li/Mg is a robust indicator of aragonite growth rate. The Li/Mg paleothermometer relationship reported previously[47,48] reflects the role of temperature in driving aragonite precipitation rate[2] (and Fig. 1f). Aragonite Li/Mg offers great potential in reconstructing $\Omega_{CM}$ in coral specimens when environmental temperature is known. The discovery of this proxy will better enable the identification of the response of marine calcifiers $\Omega_{CM}$ to environmental changes, such as rising seawater temperature and/or $pCO_2$.

## Methods
2 sets of experiments were conducted using 3 batches of seawater. In the first set of experiments, aragonite was precipitated over variable pH and $\Omega_{Ar}$ at 25 °C, with and without amino acids (Table S1). These aragonite precipitation rates are published[16]. In the second set of experiments, aragonite was precipitated over variable temperature and $\Omega_{Ar}$ (Table S2). For full details of the apparatus and methods used, see Castillo-Alvarez et al., 2024[16]. In all experiments synthetic aragonite was precipitated from artificial seawater[50] with salinity 35. The seawater was bubbled with atmospheric air to reach equilibrium and then adjusted to the required DIC/pH conditions by the addition of 0.6 M $Na_2CO_3$ (to increase DIC) and by 2 M HCl or NaOH (to control pH). The reaction vessel was supplied either with an airstream with the $[CO_2]$ adjusted to be in equilibrium with the manipulated seawater i.e. either with atmospheric air or enriched or depleted in $CO_2$ to give a composition in equilibrium with the treatment.

Seawater [DIC] was measured at the start and end of each experiment using a $CO_2$ differential, non-dispersive, infrared gas analyser (Apollo Sci-Tech; AS-C3) calibrated with a certified reference material (CRM 194, Scripps Institution of Oceanography). Seawater [DIC] error was calculated from the standard deviation of 3 to 8 replicate measurements of the sample and was 0.22% on average and always <0.7%. Between the start and end of the titration, seawater DIC varied by <5% over the first set of precipitations[16] and by ~3%, on average, over the precipitations conducted at different temperatures (Table S2). ~200 mg of an aragonite seed was added to each titration to provide a surface for aragonite growth. For the experiments over variable pH, with and without amino acids, the seed was made from a coral

skeleton[16], while for the variable temperature experiments, the seed was a synthetic aragonite[38]. The coral and synthetic seed had Brunauer Emmett Teller technique surface areas of $4.27 \pm 0.11$ ($n = 3$) and $7.00 \pm 0.20$ ($n = 3$) m$^2$ g$^{-1}$ respectively.

A Metrohm Titrando 902 pH stat titrator dosed equal volumes of 0.45 M $CaCl_2$ and $Na_2CO_3$ to maintain constant pH (and $\Omega$) in the reaction vessel during each titration. The pH of the reaction solution was monitored using a combined pH/temperature sensor (Metrohm Aquatrode PT1000). The sensor was calibrated weekly with fresh NIST (National Institute of Standards and Technology) buffers. pH was measured on the NBS (National Bureau of Standards) scale but was converted to the total scale for comparison to previous reports in the literature. $pH_{total}$ is 0.136–0.137 units lower than $pH_{NBS}$ over the pH range used here. pH drift between weeks was ≤0.003 pH units. Temperature was maintained by placing the reaction vessel in a water bath equipped with a chiller. The precipitating solution temperature was monitored at 5 s intervals within each titration using the combined pH/temperature sensor. Variations in temperature were small (±0.04 °C (1 s), on average and ±0.11°C (1 s) at most (Table S2). The titration finished when 300 mg of aragonite precipitated onto the seed and the final solid was collected by vacuum filtration onto a 0.2 μm polyether sulfone filter, rinsed with trace element grade ethanol, dried and stored in a desiccator. Each set of conditions were replicated 3 or 4 times and 1 to 3 of these replicates were analysed for geochemistry. Samples for solution [Li$^+$], [Mg$^{2+}$] and [Ca$^{2+}$] were collected at the start and end of each of the titrations collected over different temperatures. For the remaining titrations, elements were measured for each batch of seawater, at the start of select experiments (to confirm that seawater Me/Ca was not significantly affected by the addition of reagents (e.g., HCl, NaOH or amino acids) and at the end of all precipitations. Aragonite precipitation rates were calculated by normalising the rate of titrant addition to the surface area of the seed. All precipitates were confirmed as aragonite by Raman spectroscopy[46].

Solution and solid elements were determined using a Varian ICP-OES and an Agilent 7500ce ICP-MS, respectively at GEOMAR, accordingly to previous methods[13]. Seawater samples were analysed in 3 runs and details of the accuracy and precision of repeat analyses of IAPSO seawater are shown in Table S3. Procedural blanks, consisting of type 1 water filtered, stored and acidified as for seawater samples, had [Ca], [Mg] and [Li] of 0.013 mM, 0.18 mM and 0.00 μM, respectively, which, in each case, were <1 % of IAPSO seawater values. Between the start and end of each titration, seawater [Ca], [Mg] and [Li] decreased by 5, 3 and 2% respectively, on average and by 16% for all elements, at most. Mg/Ca, Li/Ca and Li/Mg changed by 3, 4 and 1% over a titration on average and by a maximum of 12, 14 and 2% respectively. The precipitates and seeds were analysed in 3 runs. The mean values for 2 reference materials, JCP-1[61] and NIST RM 8301[62], are presented in Table S4. The geochemistry of the synthetic aragonite formed in the precipitations was calculated by correcting for the seed composition, assuming the seed comprised 40% of the total solid mass.

Solution $[CO_3^{2-}]$ was calculated from measurements of [DIC] and pH at the start and end of each titration using CO2SYS v2[63] with the equilibrium constants for carbonic acid[64] and $KHSO_4$[65] and using [B] seawater[66]. Solution $\Omega_{Ar}$ was calculated from $[CO_3^{2-}]$ and $[Ca^{2+}]$ coupled with the solubility product ($K_{sp}$) of aragonite at 1 atmosphere and the precipitation temperature[67].

We use multiple linear regression analysis to identify significant influences on $D_{Mg/Ca}$, $D_{Li/Ca}$ and $D_{Li/Mg}$ and one-way ANCOVA to test for equality of means and equality of slopes in the $D_{Me/Me}$ versus log precipitation rate or $\Omega_{Ar}$ relationships between experiments conducted with and without amino acid. To identify the origin of significant differences, we conducted individual ANCOVA tests between each amino acid and the control, applying a Bonferroni correction to avoid type 1 errors. We undertook 6 ANCOVA tests (3 amino acids versus the controls for precipitation rate and $\Omega_{Ar}$) and we calculated an adjusted α value of 0.020 (i.e., the original α value, 0.05, divided by the square root of the total number of tests). We conclude that relationships are significantly different in mean (after adjusting for rate or $\Omega_{Ar}$) or slope if p ≤ 0.02.

Details of experiment conditions, seawater chemistry, aragonite chemistry and calculated distribution coefficients are in Table S1 (variable pH experiments with and without amino acids) and Table S2 (variable temperature experiments). Equations describing $D_{Me/Me}$ as a function of aragonite growth rate and pH in precipitations conducted at 25 °C in the presence of 2 mM aspartic acid, glutamic acid or glycine are summarised in Table S5.

## Estimating experimental errors

We considered the errors likely to influence our experiments. In the titrations, pH uncertainty is estimated from the maximum drift in sensor pH observed over a week (as ±0.003 units). Temperature error reflects the average temperature variation (1 s) within each precipitation (±0.04 °C). $\Omega_{Ar}$ error for each precipitation is estimated by compounding the effects of errors in measurement of [DIC], [Ca$^{2+}$], pH and temperature on $\Omega_{Ar}$.

The pH error of 0.003 affects estimated [CO$_3^{2-}$], and thereby $\Omega_{Ar}$, by 0.6%. The average error in individual [DIC] measurements (0.22%) affects estimated [CO$_3^{2-}$], and thereby $\Omega_{Ar}$, by 0.22%. The average error in seawater [Ca$^{2+}$] measurement (0.3%, Table S3) affects $\Omega_{Ar}$, by 0.3%. A temperature change of 0.04°C influences the CO$_2$ acidity constants (K*$_1$ and K*$_2$), used to calculate [CO$_3^{2-}$], by ~0.08% and 0.15% respectively[64], in combination generating a variation in [CO$_3^{2-}$], and thereby $\Omega_{Ar}$, of ~0.15%. Temperature changes of 0.04 °C influence the aragonite solubility product (K$_{sp}$) used to calculate $\Omega_{Ar}$ by <0.02%[67] and we consider this effect to be insignificant.

We calculate the $\Omega_{Ar}$ error for each precipitation by compounding the errors in [DIC] and [Ca$^{2+}$] at the start and end of each titration with the pH drift error (0.6%) and the impact of changes in CO$_2$ acidity constants (0.15%) as:

$$\Omega_{Ar}error(\%) = \sqrt{([DIC]_{start}error^2 + [DIC]_{end}error^2 + [Ca^{2+}]_{start}error^2}$$
$$\overline{+[Ca^{2+}]_{end}error^2 + pH\,error^2 + acidity\,constant\,effect^2)}$$

$$\Omega_{Ar}error(\%) = \sqrt{(0.22^2 + 0.22^2 + 0.3^2 + 0.3^2 + 0.6^2 + 0.15^2)} = 0.81\%.$$

This is equivalent to an error in $\Omega_{Ar}$ of ~0.04 at $\Omega_{Ar}$ = 5 and of ~0.16 at $\Omega_{Ar}$ = 20.

Precipitation rate is estimated by normalising the rate of titrant addition to the surface area of the seed. Replicate BET measurements of the ground coral seed used for experiments over variable pH, with and without amino acids, yield a surface area error of 2.6% (1 s[13]). Replicating BET measurements of the synthetic aragonite seed used for experiments conducted at different temperatures yields a surface area error of 2.9% (1 s). The mass of the starting seed was 200 ± 1 mg (an error of 0.5%). Compounding errors in seed mass and surface area yields an error in the estimated aragonite precipitation rate of ≤2.9%.

In calculating $D_{Me/Me}$, aragonite Me/Me is calculated by assuming that the aragonite is composed of 60% synthetic aragonite deposited during the titration and 40% seed. For the experiments over variable pH with and without amino acids, the seed was made from a coral skeleton, while for the variable temperature experiments, the seed was a synthetic aragonite. Typical errors in measurement (1 s) of aragonite Mg/Ca, Li/Ca and Li/Mg are ±0.060 mmol mol$^{-1}$, ±0.12 µmol mol$^{-1}$ and ±0.026 mmol mol$^{-1}$, respectively, based on repeat analysis of reference materials (Table S4), equivalent to 1.5%, 2.2% and 1.8%. The mass of the starting seed was 200 ± 1 mg (an error of 0.5% which we consider insignificant). Repeat analyses of replicate samples of the coral skeleton seed yield errors (1 s) in Mg/Ca, Li/Ca and Li/Mg of 0.0064 mmol mol$^{-1}$, 0.11 µmol mol$^{-1}$ and 0.023 mmol mol$^{-1}$ respectively (Table S1). Repeat analyses of the synthetic seed yield errors (1 s) in Mg/Ca, Li/Ca and Li/Mg of 0.097 mmol mol$^{-1}$, 0.12 µmol mol$^{-1}$ and 0.060 mmol mol$^{-1}$ respectively (Table S2). The seed contributes 40% of the mass of the solid collected at the end of the precipitation so variations in seed geochemistry of this magnitude could influence the final Me/Me by 40% of these values. Assuming that errors are random we estimate a total error for aragonite Mg/Ca of 0.072 mmol mol$^{-1}$ by compounding the precision of Mg/Ca of the solid (±0.060 mmol mol$^{-1}$) with the error associated with variation in Mg/Ca of the starting seed (up to ±0.039 mmol mol$^{-1}$). Similarly, we estimate errors for aragonite Li/Ca and Li/Mg of 0.13 µmol mol$^{-1}$ and 0.035 mmol mol$^{-1}$ respectively. These are equivalent to total error in aragonite Mg/Ca, Li/Ca and Li/Mg of ~1.4%, 2.4% and 3.0%, at the synthetic aragonite concentrtaions observed here (i.e., Mg/Ca, Li/Ca and Li/Mg of ~5 mmol mol$^{-1}$, ~5.5 µmol mol$^{-1}$ and ~1.2 mmol mol$^{-1}$ respectively, Tables S4 and S5).

Typical errors in measurement (1 s) of seawater Mg/Ca, Li/Ca and Li/Mg are <0.016 mol mol$^{-1}$, <0.025 mmol mol$^{-1}$ and <0.050 mmol mol$^{-1}$ or 0.3%, 1.0% and 1.0%, respectively. Compounding errors in seawater and aragonite Me/Me yields error on $D_{Mg/Ca}$, $D_{Li/Ca}$ and $D_{Li/Mg}$ of 1.4%, 2.6% and 3.1%.

## Reporting summary

Further information on research design is available in the Nature Portfolio Reporting Summary linked to this article.

## Data availability

Details of experiment conditions, seawater chemistry, aragonite chemistry and calculated distribution coefficients for this study are available in "Aragonite Li/Mg as an indicator of calcification media saturation state in marine calcifiers", Mendeley Data, https://data.mendeley.com/datasets/3dfzkxy7zm/1.

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

## Acknowledgements

Funding for this project was provided by the UK Natural Environment Research Council (NE/S001417/1). Raman analyses were supported by the EPSRC Light Element Analysis Facility Grant EP/T019298/1 and EPSRC Strategic Equipment Resource Grant EP/R023751/1 at the University of St. Andrews.

## Author contributions

N.A., A.F., K.P., R.K. and M.C. designed the study. C.C.A., N.A. and E.H. conducted the study. N.A. and C.C.A. wrote the initial draft of the paper and all authors contributed to the interpretation and final manuscript.

## Competing interests

The authors declare no competing interests.
