## [Transparent Peer Review file · Communications Earth & Environment]

Aragonite Li/Mg as an indicator of calcification media saturation state in marine calcifiers

Corresponding Author: Dr Nicola Allison

This manuscript has been previously reviewed at another Nature Portfolio journal. This document only contains reviewer comments and rebuttal letters for versions considered at Communications Earth & Environment.

Version 0:

Decision Letter:

Dear Dr Allison,

Please accept our apologies for the delay in sending a decision on your manuscript. Your manuscript titled "Aragonite Li/Mg as an indicator of calcification media Ω in marine calcifiers" has now been seen by 2 reviewers, and we include their comments at the end of this message. They find your work of interest, but some important points are raised. We are interested in the possibility of publishing your study in Communications Earth & Environment, but would like to consider your responses to these concerns and assess a revised manuscript before we make a final decision on publication.

We therefore invite you to revise and resubmit your manuscript, along with a point-by-point response that takes into account the points raised. Please highlight all changes in the manuscript text file.

Please submit your point-by-point responses as a separate file, distinct from your cover letter where you can add responses to the Editors' comments that you do not want to be made available to the reviewers. Word files are preferred. We recommend that any figures, tables or graphs that are included in the response to reviewers are also included in the main article or Supplementary Information.

Please use the following link to submit your revised manuscript, point-by-point response to the referees' comments (which should be in a separate document to any cover letter), a tracked-changes version of the manuscript (as a PDF file) and the completed checklist:

Link Redacted

We hope to receive your revised paper within six weeks; please let us know if you aren't able to submit it within this time so that we can discuss how best to proceed. If we don't hear from you, and the revision process takes significantly longer, we may close your file. In this event, we will still be happy to reconsider your paper at a later date, as long as nothing similar has been accepted for publication at Communications Earth & Environment or published elsewhere in the meantime.

Please do not hesitate to contact us if you have any questions or would like to discuss these revisions further. We look forward to seeing the revised manuscript and thank you for the opportunity to review your work.

Best regards,

Alice Drinkwater, PhD

Associate Editor
Communications Earth & Environment
Consulting Editor
Communications Sustainability

EDITORIAL POLICIES AND FORMATTING

- Behavioural and social science
- Ecological, evolutionary & environmental sciences
- Life sciences

Furthermore, please align your manuscript with our format requirements, which are summarized on the following checklist: <https://www.nature.com/documents/commsj-phys-style-formatting-checklist-article.pdf> Communications Earth & Environment formatting checklist

and also in our style and formatting guide <https://www.nature.com/documents/commsj-phys-style-formatting-guide-accept.pdf> Communications Earth & Environment formatting guide .

***** DATA:** Communications Earth & Environment endorses the principles of the Enabling FAIR data project (<http://www.copdess.org/enabling-fair-data-project/>). We ask authors to make the data that support their conclusions available in permanent, publically accessible data repositories. (Please contact the editor if you are unable to make your data available).

All Communications Earth & Environment manuscripts must include a section titled "Data Availability" at the end of the Methods section or main text (if no Methods). More information on this policy, is available at <http://www.nature.com/authors/policies/data/data-availability-statements-data-citations.pdf>.

If a community resource is unavailable, data can be submitted to generalist repositories such as <https://figshare.com/> or <http://datadryad.org/> Dryad Digital Repository. Please provide a unique identifier for the data (for example a DOI or a permanent URL) in the data availability statement, if possible. If the repository does not provide identifiers, we encourage authors to supply the search terms that will return the data. For data that have been obtained from publically available sources, please provide a URL and the specific data product name in the data availability statement. Data with a DOI should be further cited in the methods reference section.

REVIEWER COMMENTS:

Reviewer #1 (Remarks to the Author):

The manuscript "Aragonite Li/Mg as an indicator of calcification media Ω in marine calcifiers" report the use of Li/Mg ratio as a proxy of the precipitation rate in corals, this rate being itself dependent of the saturation state of the calcifying fluid.

Major Comments

Title Clarification

The current title is overly broad. Although the term "marine calcifiers" is used, the study is restricted to corals. Furthermore, only temperate and tropical corals are considered. To avoid confusion, please revise the title to accurately reflect the study's scope (e.g., "Aragonite Li/Mg as an Indicator of Calcification Media Ω in Tropical and Temperate Corals").

Experimental Basis and pH Scale Clarification

The manuscript relies on aragonite precipitation data from Castillo Alvarez et al. (2024). It is unclear whether the pH scale (e.g., pH_{NBS} vs. pH_{total}) used in the current manuscript matches that of the original dataset. This must be clearly specified. If a different scale is used, a rationale for the conversion and its implications should be provided.

Error Bars and Uncertainty Estimates

Figures 1, 2, and 3, along with corresponding data tables, do not currently show error bars. Including standard deviations or propagated uncertainties is essential for assessing data quality, trends, and variability. Please update all figures, equations and relevant tables to include appropriate error estimates.

Comparison with Existing Proxies (e.g., B/Ca)

A discussion about the advantages and limitations of using Li/Mg over B/Ca ratios to reconstruct Ω should be included. There are previous studies, particularly from McCulloch and collaborators, on using B/Ca and $\delta^{11}\text{B}$ as proxies for calcifying fluid chemistry. The manuscript would benefit from citing and discussing these studies to contextualize the contribution of the Li/Mg proxy.

L45-47: Since specific decoupled pH/DIC experiments were not conducted, please provide more detail on how this separation is addressed analytically or conceptually within your framework.

L119-129: Please provide a comparison between your experimental data and natural coral partitioning behavior

Figure 1: add the error bars. Please specify the salinity.

Figure 2: Add error bars. Include ranges from the literature for both experimental aragonite and natural coral data for context.

L119-129: again show how your data compared to data from natural corals.

L161: This section is not about paleoenvironmental analysis but only on modern corals. Please modify.

L174: add the errors on the coefficients of the equation

L179: The word "and" is incorrectly formatted as a subscript; please correct.

L195: Specify that it is also for temperatures below 30°C

Figure 3: please add the error bars

L207: please add the errors to the coefficients of the equation

L223: To support this proxy application, a test would be to use Li/Mg ratios from cultured coral skeletons or aragonite experiments to predict growth rates or Ω and then compare these predictions to independent measurements or published values. This would greatly enhance the credibility and applicability of the proxy.

Conclusion

This manuscript addresses a promising proxy system (Li/Mg) for assessing coral calcification dynamics and Ω . However, significant improvements are needed in the clarity of the experimental framework, error reporting, and contextualization of the proxy within the broader literature. Once these major concerns are addressed and the manuscript is thoroughly revised, it may merit reconsideration for publication.

Reviewer #2 (Remarks to the Author):

See the attached file.

** Visit Nature Portfolio's author and referees' website at www.nature.com/authors for information about policies, services and author benefits**

Communications Earth & Environment is committed to improving transparency in authorship. As part of our efforts in this direction, we are now requesting that all authors identified as 'corresponding author' create and link their Open Researcher and Contributor Identifier (ORCID) with their account on the Manuscript Tracking System prior to acceptance. ORCID helps the scientific community achieve unambiguous attribution of all scholarly contributions. You can create and link your ORCID from the home page of the Manuscript Tracking System by clicking on 'Modify my Springer Nature account' and following the instructions in the link below. Please also inform all co-authors that they can add their ORCIDs to their accounts and that they must do so prior to acceptance.

Version 1:

Decision Letter:

Dear Dr Allison,

Your manuscript titled "Aragonite Li/Mg as an indicator of calcification media Ω in marine calcifiers" has now been seen by our reviewers, whose comments appear below. In light of their advice we are delighted to say that we are happy, in principle, to publish a suitably revised version in Communications Earth & Environment.

We therefore invite you to revise your paper one last time to address the remaining concerns of our reviewers. At the same time we ask that you edit your manuscript to comply with our format requirements and to maximise the accessibility and therefore the impact of your work.

EDITORIAL REQUESTS:

****Please take care to match our formatting and policy requirements. We will check revised manuscript and return manuscripts that do not comply. Such requests will lead to delays. ****

SUBMISSION INFORMATION:

OPEN ACCESS:

Communications Earth & Environment is a fully open access journal. Articles are made freely accessible on publication. For further information about article processing charges, open access funding, and advice and support from Nature Portfolio, please visit <https://www.nature.com/commsenv/open-access>

Link Redacted

Best regards,

Alice Drinkwater, PhD
Associate Editor
Communications Earth & Environment
Consulting Editor
Communications Sustainability

REVIEWERS' COMMENTS:

Reviewer #1 (Remarks to the Author):

I appreciate the authors' efforts in revising the manuscript and providing thoughtful responses to the reviewers' comments. While the manuscript has improved, I believe there are still a few aspects that could benefit from further developments.

1) Comparison with previous studies (lines 40 to 43)

Please develop this part further. If there is a change in the $\delta^{11}\text{B}$ signature of the calcifying fluid as suggested in Allison et al (2023), how can one expect that Li/Mg remains unaffected? I don't say that this ratio is indeed affected but this point requires a more detailed discussion. Note also that in some studies of Venn et al., they found a strong agreement between $\delta^{11}\text{B}$ and measured pH, which suggests that $\delta^{11}\text{B}$ reflects the pH of the extracellular calcifying medium (ECM). In addition, note that for B/Ca, the no sensitivity appears to be limited to pH values below ~ 8.41 as stated in Alvarez et al 2024, while in the ECM the pH could be above to this threshold (e.g., Venn et al., 2025).

2) Estimation of growth rate

It may be worth considering the study by Brahma et al. (2012), as it could offer an independent way to estimate coral growth rates and potentially add further support to the present analysis.

Reviewer #2 (Remarks to the Author):

The authors have thoroughly addressed all concerns from the first round of review, and I recommend the manuscript be accepted without further revision

** Visit Nature Portfolio's author and referees' website at www.nature.com/authors for information about policies, services and author benefits**

Response to reviewers - Submission COMMSENV-25-2494-T

We thank the editor and reviewers for taking the time to read the paper and make constructive comments that have allowed us to improve the paper. We have pasted the reviewer's comments below in bold typeface. We've divided the reviewer text to deal with each point in turn. Our responses to the reviewer's comments are in regular text. Line numbers refer to the revised and highlighted version of the manuscript uploaded as a pdf.

Reviewer #1 (Remarks to the Author):

Major Comments

Title Clarification

The current title is overly broad. Although the term "marine calcifiers" is used, the study is restricted to corals. Furthermore, only temperate and tropical corals are considered. To avoid confusion, please revise the title to accurately reflect the study's scope (e.g., "Aragonite Li/Mg as an Indicator of Calcification Media Ω in Tropical and Temperate Corals").

We have kept the original title. The paper presents data on the effects of temperature and growth rate on the Li/Mg of synthetic aragonite. These data are relevant to all aragonitic marine calcifiers. In Figure 3a we reproduce the aragonite Li/Mg dataset for aragonitic corals and foraminifera published by Williams et al., 2023 and we interpret this in relation to our data (Figure 3b).

Experimental Basis and pH Scale Clarification

The manuscript relies on aragonite precipitation data from Castillo Alvarez et al. (2024). It is unclear whether the pH scale (e.g., pH_{NBS} vs. pH_{total}) used in the current manuscript matches that of the original dataset. This must be clearly specified. If a different scale is used, a rationale for the conversion and its implications should be provided.

This paper reports Mg and Li analyses of precipitations reported by Castillo Alvarez et al. (2024) and of a new suite of precipitations conducted for this research. We have expanded the methods section to make this clearer (lines 454-457). All experiments are presented on the total pH scale as stated in the axis label of Figure 1b and the legends for Tables 1, S4 and S5. We have added a sentence to the method to explain that all pH data are recorded on the NBS scale but are converted to the total scale (lines 480-484).

Error Bars and Uncertainty Estimates

Figures 1, 2, and 3, along with corresponding data tables, do not currently show error bars. Including standard deviations or propagated uncertainties is essential for assessing data quality, trends, and variability. Please update all figures, equations and relevant tables to include appropriate error estimates.

We have added a section to the methods to discuss the uncertainties in our estimates of seawater pH, Ω_{Ar} , precipitation rate and $D_{Mg/Ca}$, $D_{Li/Ca}$ and $D_{Li/Mg}$ (lines 531-586). We have expanded Table S2 (details of the precipitations conducted over variable temperature) to include the start, end and mean DIC of each titration and to include measured seawater temperature and temperature error. We have expanded Tables S3 (detail of the analyses of the standard seawater analysed with the seawater samples) and Table S4 (detail of the aragonite reference materials analysed with the aragonite samples) to show the results of all replicates to demonstrate the precision of repeat analyses. These errors are discussed in the error section. We have added detail to the legends of Figures 1 and 2 to summarise these errors and to state that all errors are smaller than the sizes of the symbols used. All the tables and equations state the error associated with analyses.

Comparison with Existing Proxies (e.g., B/Ca)

A discussion about the advantages and limitations of using Li/Mg over B/Ca ratios to reconstruct Ω should be included. There are previous studies, particularly from McCulloch and collaborators, on using B/Ca and $\delta^{11}B$ as proxies for calcifying fluid chemistry. The manuscript would benefit from citing and discussing these studies to contextualize the contribution of the Li/Mg proxy.

We have added 2 sentences to the introduction to highlight that aragonite B/Ca has been used to infer calcification media [CO_3^{2-}] but we note that recent advances suggest that these proxies may not be reliable (lines 40-43).

L45-47: Since specific decoupled pH/DIC experiments were not conducted, please provide more detail on how this separation is addressed analytically or conceptually within your framework.

Specific decoupled pH/DIC experiments were conducted to separate pH and DIC/omega effects. We have rewritten this sentence to be clearer. We now state '[CO_3^{2-}] and pH covary between precipitations at atmospheric CO_2 (ref. 13), so we conduct experiments under different CO_2 atmospheres to deconvolve the influences of pH and [CO_3^{2-}] on Mg and Li partitioning (line 50).

L119-129: Please provide a comparison between your experimental data and natural coral partitioning behavior.

We have added a paragraph to the discussion to make a broad comparison of the experiment data and element partitioning in biogenic aragonite (lines 238-254). We state:

'To broadly compare element partitioning between the synthetic aragonite samples and biogenic aragonite, we calculate $D_{\text{Mg/Ca}}$, $D_{\text{Li/Ca}}$ and $D_{\text{Li/Mg}}$ for the *Porites* spp. corals skeletons illustrated in Fig. 3c. We use the skeletal Mg/Ca and Li/Mg data summarised in Williams et al. (2023)³³ and assume that calcification media [Mg], [Li] and [Ca] approximates that of seawater i.e. $52.7 \text{ mmol kg}^{-1}$, $25.9 \text{ } \mu\text{mol kg}^{-1}$ and $10.3 \text{ mmol kg}^{-1}$ (ref 50). This yields coral skeletal $D_{\text{Mg/Ca}} = 0.79 \pm 0.08 \times 10^{-3}$ ($n = 38$), $D_{\text{Li/Ca}} = 2.6 \pm 0.2 \times 10^{-3}$ ($n=38$) and $D_{\text{Li/Mg}} = 3.1 \pm 0.4$ ($n = 211$). These values are comparable to the distribution coefficients observed in the synthetic aragonite precipitations reported here (Fig. 2), although we note that the synthetic experiments span a broad range of precipitation rates. We know of no independent estimates of aragonite precipitation rate in coral skeletons. Coral calcification rates are usually reported normalised to the surface area of the coral colony but aragonite deposition occurs over the existing skeleton in contact with the coral tissue and this represents a much larger surface area⁵⁹. At this time a more detailed comparison of distribution coefficients between biogenic and synthetic aragonite is not possible'.

Figure 1: add the error bars. Please specify the salinity. All experiments were conducted at salinity = 35. We have added a sentence to the main article to state this (lines 56 and 114). We have added a section to the methods to discuss the uncertainties in our estimates of seawater pH, Ω_{Ar} and precipitation rate (lines 238-254). We have added detail to the legends of Figure 1 to summarise these errors and to state that all errors are smaller than the sizes of the symbols.

Figure 2: Add error bars. Include ranges from the literature for both experimental aragonite and natural coral data for context. We have added a section to the methods to discuss the uncertainties in our estimates of seawater pH, Ω_{Ar} and precipitation rate (see above). We have added detail to the legends of Figure 2 to summarise these errors and to state that all errors are smaller than the sizes of the symbols.

L119-129: again show how your data compared to data from natural corals. We have covered this in the point above.

L161: This section is not about paleoenvironmental analysis but only on modern corals. Please modify. We have changed the title of this section to 'Applications to biogenic aragonite' (line 163).

L174: add the errors on the coefficients of the equation. Here we rearrange equation 8 from Table 1. The errors on the coefficients are stated in Table 1 and we do not repeat them here (line 188).

L179: The word “and” is incorrectly formatted as a subscript; please correct. Done (line 206).

L195: Specify that it is also for temperatures below 30°C. We have altered this in the legend to Figure 3 (line 195).

Figure 3: please add the error bars. Here we have reproduced the dataset from Williams et al., 20203. We do not have error bars for this data. The error of aragonite Mg/Ca, Li/Ca and Li/Mg analysis is typically small (see Table S3) and smaller than the symbols used in our figures.

L207: please add the errors to the coefficients of the equation Here we rearrange equation 4 from Table 1. The errors on the coefficients are stated in Table 1 and we do not repeat them here (Line 221).

L223: To support this proxy application, a test would be to use Li/Mg ratios from cultured coral skeletons or aragonite experiments to predict growth rates or Ω and then compare these predictions to independent measurements or published values. This would greatly enhance the credibility and applicability of the proxy. Yes, but there are no independent estimates of aragonite precipitation rate in biogenic aragonite samples. We now discuss this in the discussion. Specifically we state that ‘We know of no independent estimates of aragonite precipitation rate in coral skeletons. Coral calcification rates are usually reported normalised to the surface area of the coral colony but aragonite deposition occurs over the existing skeleton in contact with the coral tissue and this represents a much larger surface area⁵⁹. At this time a more detailed comparison of distribution coefficients between biogenic and synthetic aragonite is not possible.’ (lines 249-254).

Conclusion

This manuscript addresses a promising proxy system (Li/Mg) for assessing coral calcification dynamics and Ω . However, significant improvements are needed in the clarity of the experimental framework, error reporting, and contextualization of the proxy within the broader literature. Once these major concerns are addressed and the manuscript is thoroughly revised, it may merit reconsideration for publication.

Reviewer 2.

This manuscript described a study of Li/Mg partition behavior during the aragonite growth process. The authors designed experiments to evaluate the effects of pH, growth rate, temperature, aragonite saturation state (Ω), and amino acids. The experiments were systematically and carefully performed, and the data are properly interpreted. Based on the results, this study proposed a new Li/Mg proxy to reconstruct the Ω of biological calcification media and applied it to several modern corals. In general, this study is well conceived and could be a nice piece of work with novelty. However, there are some issues that must be addressed before this manuscript can be considered for publication. Blow are my comments to the manuscript:

(1) Non-classical crystallization pathways. The application of the Li/Mg proxy may be limited by the non-classical crystallization mechanisms often observed in biogenic aragonite. Numerous studies have reported that aragonite in corals and bivalves can form through the transformation of amorphous calcium carbonate (ACC), rather than by direct ion-by-ion growth. The partitioning behavior of trace elements during ACC formation and transformation could differ significantly from classical growth processes, potentially affecting the reliability of the Li/Mg proxy in natural systems.

In coral aragonite there is evidence for both ACC transformation and ion-by-ion growth (Sun et al., 2019). We have added a sentence to highlight that aragonite formation via ACC may influence element partitioning (line 80-81) and to indicate in the discussion that further work is required to identify if ACC occurs as a precursor phase during these experiments and to resolve how ACC transformation affects element partitioning (line 239-241).

(2) Differences in calcification media composition. Another concern is that the chemical composition of the calcifying fluid in marine organisms often differs from that of ambient seawater. Biological processes can selectively concentrate or exclude specific ions, leading to elemental ratios in the calcifying fluid that are not representative of seawater. Consequently, Li/Mg ratios in biogenic aragonite may not directly reflect seawater chemistry, which complicates the application of this proxy in natural settings.

Agreed. We have added a paragraph to the discussion to compare element partitioning between the synthetic aragonite samples produced here and coral aragonite (lines 238-254). We observe broad agreement in the distribution coefficients between the synthetic samples and coral skeletons but as we explain the precipitation rates of biogenic aragonite samples are unknown so it is difficult to make a meaningful comparison here.

(3) Lack of contextualization with previous proxies. In the introduction section, the authors don't sufficiently review prior work on Me/Ca (where Me is trace metal) proxies for reconstructing calcification media chemistry. As a result, the novelty and specific advantages of the proposed Li/Mg proxy are not clearly established. For instance, both Li/Ca and Mg/Ca have been used in previous studies for similar purposes. The authors should clarify why Li/Mg offers an improvement over these individual ratios or other existing proxies.

We already cite previous studies on Li/Ca, Mg/Ca and Li/Mg in aragonite (lines 68-69). We have added text to the introduction to summarise the previous use of B/Ca for as an indicator of calcification media chemistry (lines 40-43).

(4) Limited comparison with previous studies. The discussion section does not fully explore how the new experimental data compare with existing literature. In particular, the authors report significantly higher D_{Li} and D_{Mg} values than those in previous studies, yet do not provide a clear explanation for this discrepancy, even though similar trends between D values and growth rate are observed. A more thorough comparison and discussion would strengthen the interpretation and contextualization of the results.

We have now added a comparison of the distribution coefficients calculated from the synthetic aragonite precipitations conducted here and coefficients calculated for corals. These values are comparable (lines 238-254). We already plot our coefficients estimates in comparison with estimates from other studies (Fig. S3). We note that these previous studies used simple solutions with low ionic strength (not seawater) and [Li] that is more than x 500 higher than that of seawater (lines 131-133). Further work is required to determine how Li partitioning is influenced by [Li] (lines 137).

Specific comments:

L32-34: The precipitation rate could be also affected by the presence of foreign ions, such as Mg²⁺, SO₄²⁻, and PO₄³⁻-etc.

We have expanded this sentence to make this clear. We now state 'Inorganic precipitation rates of CaCO₃ reflect seawater CaCO₃ saturation state, Ω , which is a function of seawater [Ca²⁺] and [CO₃²⁻] (ref. 2) and the presence of other ions e.g. Mg²⁺, SO₄²⁻ and PO₄³⁻ (line 34).

L35-37: Yes, the increasing CO₃²⁻ concentrations make the formation of ACC possible.

We have been unable to produce ACC under the conditions used here in our other experiments (Evans et al., 2020).

L40-50: Before accepting a new proxy, I'd like to know the disadvantages of the previous ones, or the inconsistencies between different studies when using the traditional proxies.

We have added 2 sentences to explain this (lines 40-43).

L78-79: "...smaller ionic radii than Ca²⁺" and update the reference format

We have altered this to read 'Both Mg²⁺ and Li⁺ have smaller ionic radii than Ca²⁺ (ref. 25).' The reference format is as requested by the journal (line 85).

L79-81: What about incorporation of ions as fluid inclusion? We know of no evidence that significant amounts of these elements are incorporated in fluid inclusions in aragonite.

L81-83: Why? The pH effect can be observed in calcite. We have rewritten this as 'Although there is evidence of LiHCO₃ incorporation into calcite^{17,18}, we observe no effect of pH (Table 1) or fluid [HCO₃⁻] (Fig. S2) on D_{Li/Ca} to support this hypothesis in aragonite.' (Line 89). We note that 'relatively large variations occur in D_{Li/Ca} between different pH treatments at high aragonite growth rates (Fig. 2b) which warrant future investigation' (lines 104-106).

L107-108: That's interesting. We expect a positive relationship between seawater pH and Ω_{Ar} at constant [CO₂].

L119-121: Why? We have not hypothesised here. We note that 'these previous studies used simple solutions with low ionic strength (not seawater) and [Li] that is more than x 500 higher than that of seawater (Lines 131-133) Further work is required to determine how Li partitioning is influenced by [Li] (line 137).

L141: Update the reference format. Done (line 153).

L131-149: Is it possible that the amino acid can preferentially form ion-complex with specific ions, thus changing the free ion ratios in the solution and resulting in different ion ratios in aragonite.

Yes, we note that 'Amino acids complex Mg²⁺ and Ca²⁺ in solution³⁷' (line 152) and this 'could alter the relative rates of trace element and Ca²⁺ adsorption to the mineral surface' (line 153). However 'the seawater [amino acid] tested here (2 mM) far exceeds that likely to occur at organism calcification sites. Intra-crystalline [aspartic acid] is 0.5 to 1.5 nmol mg⁻¹ in coral skeletons^{32,33} and ≤1 nmol mg⁻¹ in mollusc shells³⁹, but is >13 nmol mg⁻¹ in synthetic aragonite precipitated under the conditions used here⁴⁰. *In vitro* precipitations with 100 μM aspartic acid produce aragonite which has a comparable [aspartic acid] (0.8 nmol mg⁻¹, ref 40), to biominerals. Aragonite precipitation rates are reduced by ~12% by 100 μM aspartic acid⁴¹ and any influence of this on aragonite Li/Mg will be small (Fig 2i)' (line 155-161).

References

Evans D, Gray WR, Rae J, Greenop R, Webb PB, Penkman K, Kroger R & Allison N, Trace element incorporation into amorphous calcium carbonate (ACC) precipitated from seawater, *Geochimica et Cosmochimica Acta*, 290, 293-311, 2020.

Sun, C.Y., Stifler, C.A., Chopdekar, R.v., Schmidt, C.A., Parida, G., Schoeppler, V., Fordyce, B.I., Brau, J.H., Mass, T., Tambutté, S., Gilbert P.U.P.A., 2020. From particle attachment to space-filling coral skeletons, *Proc. Natl. Acad. Sci. USA*, 117, 30159–30170.

Williams, TJ et al. (2023): A global database of measured values of Li/Mg, Mg/Ca, Sr/Ca, Ba/Ca, U/Ca and Sr-U for coral and coralline algae paleoenvironment calibrations [dataset bundled publication]. PANGAEA, <https://doi.org/10.1594/PANGAEA.962897>

Response to reviewers - Submission COMMSENV-25-2494-T

We thank the editor and reviewers for taking the time to read the paper and make constructive comments that have allowed us to improve the paper. We have pasted the reviewer's comments below in bold typeface. We've divided the reviewer text to deal with each point in turn. Our responses to the reviewer's comments are in regular text. Line numbers refer to the revised version of the manuscript uploaded as a pdf.

REVIEWERS' COMMENTS:

Reviewer #1 (Remarks to the Author):

I appreciate the authors' efforts in revising the manuscript and providing thoughtful responses to the reviewers' comments. While the manuscript has improved, I believe there are still a few aspects that could benefit from further developments.

1) Comparison with previous studies (lines 40 to 43)

Please develop this part further. If there is a change in the $\delta^{11}\text{B}$ signature of the calcifying fluid as suggested in Allison et al (2023), how can one expect that Li/Mg remains unaffected? I don't say that this ratio is indeed affected but this point requires a more detailed discussion. Note also that in some studies of Venn et al., they found a strong agreement between $\delta^{11}\text{B}$ and measured pH, which suggests that $\delta^{11}\text{B}$ reflects the pH of the extracellular calcifying medium (ECM). In addition, note that for B/Ca, the no sensitivity appears to be limited to pH values below ~ 8.41 as stated in Alvarez et al 2024, while in the ECM the pH could be above to this threshold (e.g., Venn et al., 2025).

We have revised this section to read :

"However, a recent study indicates that aragonite B/Ca is not influenced by precipitating fluid [CO_3^{2-}] at the pH and DIC conditions of tropical coral calcification sites¹³, while $\delta^{11}\text{B}$ estimates of coral calcification media pH are substantially higher than estimates from the pH sensitive dye SNARF-1 in the extracellular calcification site¹⁴." (lines 42-45)

Castillo Alvarez (2024) shows that $\text{B}(\text{OH})_4^-$ and CO_3^{2-} do not compete at pH(total scale) of 8.41 and [CO_3^{2-}] up to $1200 \mu\text{mol kg}^{-1}$. Direct measurements indicates that the pH_{total} of the extracellular calcification media of corals cultured at present day pCO_2 is ~ 8.3 to ~ 8.6 (Venn et al., 2011, 2013; Sevilgen et al, 2019) while [CO_3^{2-}] is $400\text{-}870 \mu\text{mol kg}^{-1}$ (Sevilgen et al, 2019). Under these conditions, $\text{B}(\text{OH})_4^-$ and CO_3^{2-} do not compete (Castillo Alvarez, 2024, Figure 2b).

2) Estimation of growth rate

It may be worth considering the study by Brahmi et al. (2012), as it could offer an independent way to estimate coral growth rates and potentially add further support to the present analysis.

We have a reference to this paper at the end of the discussion (lines 207-208). We note:

"Coral biomineralisation can also be measured as skeletal extension but this is highly variable within individual skeletons⁶⁰."

Reviewer #2 (Remarks to the Author):

The authors have thoroughly addressed all concerns from the first round of review, and I recommend the manuscript be accepted without further revision

References

Castillo Alvarez, C. et al. B(OH)_4^- and CO_3^{2-} do not compete for incorporation into aragonite in synthetic precipitations at pH_{total} 8.20 and 8.41 but do compete at pH_{total} 8.59. *Geochim. Cosmochim. Acta* **379** 39-52 (2024).

Sevilgen DS, et al. Full *in vivo* characterization of carbonate chemistry at the site of calcification in corals. *Science Adv.* 5 eaau7447 (2019).

Venn, A. A. et al. Impact of seawater acidification on pH at the tissue–skeleton interface and calcification in reef corals. *Proc. Natl. Acad. Sci. USA.* 110 1634-9 (2013).

Venn A, Tambutté E, Holcomb M, Allemand D, Tambutté S. Live tissue imaging shows reef corals elevate pH under their calcifying tissue relative to seawater. *PloS one.* 2011 May 27;6(5):e20013.

This manuscript described a study of Li/Mg partition behavior during the aragonite growth process. The authors designed experiments to evaluate the effects of pH, growth rate, temperature, aragonite saturation state (Ω), and amino acids. The experiments were systematically and carefully performed, and the data are properly interpreted. Based on the results, this study proposed a new Li/Mg proxy to reconstruct the Ω of biological calcification media and applied it to several modern corals. In general, this study is well conceived and could be a nice piece of work with novelty. However, there are some issues that must be addressed before this manuscript can be considered for publication. Below are my comments to the manuscript:

- (1) Non-classical crystallization pathways. The application of the Li/Mg proxy may be limited by the non-classical crystallization mechanisms often observed in biogenic aragonite. Numerous studies have reported that aragonite in corals and bivalves can form through the transformation of amorphous calcium carbonate (ACC), rather than by direct ion-by-ion growth. The partitioning behavior of trace elements during ACC formation and transformation could differ significantly from classical growth processes, potentially affecting the reliability of the Li/Mg proxy in natural systems.
- (2) Differences in calcification media composition. Another concern is that the chemical composition of the calcifying fluid in marine organisms often differs from that of ambient seawater. Biological processes can selectively concentrate or exclude specific ions, leading to elemental ratios in the calcifying fluid that are not representative of seawater. Consequently, Li/Mg ratios in biogenic aragonite may not directly reflect seawater chemistry, which complicates the application of this proxy in natural settings.
- (3) Lack of contextualization with previous proxies. In the introduction section, the authors don't sufficiently review prior work on Me/Ca (where Me is trace metal) proxies for reconstructing calcification media chemistry. As a result, the novelty and specific advantages of the proposed Li/Mg proxy are not clearly established. For instance, both Li/Ca and Mg/Ca have been used in previous studies for similar purposes. The authors should clarify why Li/Mg offers an improvement over these individual ratios or other existing proxies.
- (4) Limited comparison with previous studies. The discussion section does not fully explore how the new experimental data compare with existing literature. In particular, the authors report significantly higher D_{Li} and D_{Mg} values than those in previous studies, yet do not provide a clear explanation for this discrepancy, even though similar trends between D values and growth rate are observed. A more thorough comparison and discussion would strengthen the interpretation and contextualization of the results.

Specific comments:

L32-34: The precipitation rate could be also affected by the presence of foreign ions, such as Mg^{2+} , SO_4^{2-} , and PO_4^{3-} etc.

L35-37: Yes, the increasing CO_3^{2-} concentrations make the formation of ACC possible.

L40-50: Before accepting a new proxy, I'd like to know the disadvantages of the previous ones, or the inconsistencies between different studies when using the traditional proxies.

L78-79: "...smaller ionic radii than Ca^{2+} " and update the reference format

L79-81: What about incorporation of ions as fluid inclusion?

L81-83: Why? The pH effect can be observed in calcite

L107-108: That's interesting

L119-121: Why?

L141: Update the reference format

L131-149: Is it possible that the amino acid can preferentially form ion-complex with specific ions, thus changing the free ion ratios in the solution and resulting in different ion ratios in aragonite?